# Carrageenan-Based Pickering Emulsion Gels Stabilized by Xanthan Gum/Lysozyme Nanoparticle: Microstructure, Rheological, and Texture Perspective

**DOI:** 10.3390/foods11233757

**Published:** 2022-11-22

**Authors:** Tianzhen Xiong, Haomin Sun, Ziyi Niu, Wei Xu, Zhifan Li, Yawen He, Denglin Luo, Wenjie Xi, Jingjing Wei, Chunlan Zhang

**Affiliations:** 1College of Life Science, Xinyang Normal University, Xinyang 464000, China; 2College of Food and Bioengineering, Henan University of Science and Technology, Luoyang 471023, China; 3College of Food Science and Engineering, Tarim University, Alar 843300, China

**Keywords:** Pickering emulsion gels, kappa carrageenan, rheological properties, microstructure

## Abstract

In this study, Pickering emulsion gels were prepared by the self-gel method based on kappa carrageenan (kC). The effects of particle stabilizers and polysaccharide concentrations on the microstructure, rheological characteristics, and texture of Pickering emulsion gels stabilized by xanthan gum/lysozyme nanoparticles (XG/Ly NPs) with kC were discussed. The viscoelasticity of Pickering emulsion gels increased significantly with the increase of kC and XG/Ly NPs. The results of temperature sweep showed that the gel formation mainly depended on the kC addition. The XG/Ly NPs addition could accelerate the formation of Pickering emulsion gels and increase its melting temperature (T_melt_), which is helpful to improve the thermal stability of emulsion gels. Cryo-scanning electron microscope (Cryo-SEM) images revealed that Pickering emulsion gel has a porous network structure, and the oil droplets were well wrapped in the pores. The hardness increased significantly with the increase of XG/Ly NPs and kC. In particular, the Pickering emulsion gel hardness was up to 2.9 Newton (N) when the concentration of kC and XG/Ly NPs were 2%. The results showed that self-gelling polysaccharides, such as kC, could construct and regulate the structure and characteristics of Pickering emulsion gel. This study provides theoretical support for potential new applications of emulsion gels as functional colloids and delivery systems in the food industry.

## 1. Introduction

Emulsions have been widely applied and attracted much attention in the food, nutrition, medicine, and cosmetics fields [1]. Compared with traditional emulsion, Pickering emulsion stabilized by ultrafine solid particles has the advantages of reducing toxicity and improving emulsion stability [2,3]. As a typical particle stabilizer, protein/polysaccharide systems are attractive in food, cosmetics, and medicine due to their distinct advantages such as the natural ingredients, good emulsifying properties, and stability [4,5]. According to our previous studies, XG/Ly NPs are formed by self-assembly with good interfacial activity and emulsifying performance. Pickering emulsions stabilized by XG/Ly NPs were unstable in manufacturing unit for droplet coalescence and Ostwald ripening. Therefore, different approaches have been proposed to improve the current situation, such as a polysaccharide addition and increasing particle stabilizer concentration [6]. Comparatively, Pickering emulsion gels provide excellent stability due to their gel-like network and solid-like properties that protect oil droplets from oxidation [7,8].

Pickering emulsion gel is a two-phase system in which one phase dispersed in another, where the microstructures and viscoelastic properties can be tailored. It combines the virtues of both Pickering emulsions and gels [9,10,11]. Compared with Pickering emulsions, it is more stable against droplet coalescence, Ostwald ripening, and phase separation [12,13]. The properties of Pickering emulsion gels are strongly influenced by inter-droplet interactions [14]. Engineering inter-droplet interactions at molecular scale are essential to create desired Pickering emulsion gel systems. Commonly formed by filling gel matrix in continuous phase or dispersed phase, it forms a spatial network, to result in a semisolid texture. The simple method of gel formation generally uses easy-gelatinized protein and polysaccharide [15,16,17]. For example, Li provided a simple method to construct Pickering emulsion gels based on chitosan hydrochloride-carboxymethyl starch nanogels stabilized Pickering emulsions coupled curdlan gelation. It offered a new strategy for the design of low-fat food [18]. Polysaccharides, including xanthan gum, konjac glucomannan, and carrageenan, are used in Pickering emulsion gels as regulator. Among them, kappa carrageenan (kC) is excellent as a thickener and gelling agent, which is commonly used to alter viscoelasticity [19].

Carrageenan (kappa) as a linear polysaccharide is composed of D-galactose and 3, 6-dehydrated D-galactose with one sulfate groups per repeating disaccharide [20]. The kC solution (≥1%) can spontaneously form a gel at room temperature after heating at 70 °C for 20 min. At this time, the stable cross-linking region formed between kC molecules constituting a spatial three-dimensional network structure. This structure of kC could cooperate with other polysaccharides, such as konjac glucomannan and xanthan gum, to improve the elasticity and water retention of the gel [21]. It can also improve the stability of zein nanoparticles at different pH values. The approach was therefore proposed based on the formulation of Pickering emulsion gel system that exhibits self-standing viscoelastic properties using direct emulsification of all ingredients [22].

In this paper, the transformation of the Pickering emulsion system is regulated from sol to gel state based on self-gelation of kC. The effects of kC and XG/Ly NPs concentration on the structure formation and rheological properties of Pickering emulsion gel are investigated. The results will broaden the application of polysaccharides represented by kC in the construction of new gel-related food structures transformed from sol to gel state.

## 2. Materials and Methods

### 2.1. Materials

Lysozyme (Ly, Mw = 14.3 kDa) from chicken egg white, xanthan gum (XG), and Nile Red were provided by Sinopharm Chemical Reagent Co., Ltd. (Shanghai, China). Kappa carrageenan (kC) was purchased from Shanghai Aladdin Biochemical Technology Co., Ltd. (Shanghai, China). Tea seed oil (saturated fatty acid 10%, monounsaturated fatty acid 80%, and polyunsaturated fatty acids 10%) was purchased from a local supermarket which produced by Luda Green organic camellia oil Co. Ltd. (Xinyang, China). Other chemicals were of analytical grade and used without further purification. All solutions in the experiments were prepared by ultrapure water through Millipore Milli-Q water purification.

### 2.2. Preparation of Complex Solution of XG/Ly NPs and kC

XG/Ly NPs were prepared as described previously with minor modifications [23]. In brief, XG and Ly mixture solutions were first prepared with concentration 1.0 mg/mL. Then, XG solution was added into Ly solution and the mixture was adjusted to pH 11.8. The mixture was heated at 80 °C for 15 min and cooled to room temperature. The pH of the mixture was adjusted to 7.0 for 24 h dialysis against deionized water. XG/Ly nanoparticles were obtained by filtration with a membrane of 80 μm to remove impurities. kC was dispersed in XG/Ly NPs solution and heated by stirring in a water bath at 70 °C for 20 min until the kC was completely dissolved. It was stirred until it was cooled to room temperature. XG/Ly NPs would be evaluated at concentrations of 0.5%, 1%, and 2%. Additionally, kC would be independently varied using the same concentration values.

### 2.3. Preparation of Pickering Emulsion Gel

Pickering emulsion gel was prepared using mixture solutions of XG/Ly NPs and kC as an emulsifier, tea oil as oil phase. The oil/water volume fraction was 20%. The emulsions were homogenized for 3 min in ultra-Turrax T25 homogenizer at 18,000 rpm. Subsequently, Pickering emulsion gel was formed after being placed in a refrigerator at 4 °C for 12 h.

### 2.4. Characterization of Pickering Emulsion Gels

#### 2.4.1. Rheological Analysis

Discovery HR-2 rheometer equipped with a 20 mm diameter parallel plate was used to monitor the gel evolution process of Pickering emulsion, and the following three modes were utilized for testing. Temperature sweep measurements: the freshly prepared Pickering emulsion gel was immediately transferred to the center of the sample table preset at 80 °C. The gap between the two parallel plates was set to 1000 μm. Subsequently, the temperature sweep was performed at the programmed temperature from 80 to 20 °C with a cooling rate of 2 °C/min, the strain range being 1% and the frequency 1 Hz. When the sample temperature dropped to 20 °C, another temperature scanning procedure was performed at the programmed temperature from 20 to 95 °C to evaluate the thermal stability of emulsion gel. The temperature corresponding to the intersection of storage modulus (G′) and loss modulus (G″) was defined as gelation (T_gel_) or gel melting point (T_melt_) [24].Time sweep measurements: to evaluate the time dependence of the gelation process of Pickering emulsion, the time scanning program at three fixed temperatures (40 °C, 30 °C, and 20 °C, respectively) was run for 1 h at 0.5% strain and 1 Hz frequency. Frequency sweep measurements: when Pickering emulsion gel cool to 20 °C at a cooling rate of 1 °C/min on a rheometer, frequency scanning was carried out at a fixed strain of 1% (within LVR) in the frequency range of 1–100 Hz.

#### 2.4.2. Gel Microstructure

Optical microscope: The micromorphology of Pickering emulsion gels was visualized through an optical microscope (Leica, Feasterville, PA, USA) with a 40× objective. Freshly prepared Pickering emulsion gels with different preparation parameters were selected and placed on the glass slides. The images were obtained using a microscope. The size distribution was calculated using Image J software.Confocal laser scanning microscopy (CLSM): Confocal laser scanning microscopy (CLSM, Leica TCS SP8, Germany) was used to observe the microscopic morphology of Pickering emulsion gels to further understand the distribution of the dispersed phase and the continuous phase. The excitation wavelengths of Nile Red were 488 nm. Before the measurement, the sample was dyed with mixed dye formed by dissolving 0.1% Nile Red in 1, 2-propanediol. Then, 40 μL of the mixed dye solution was added to 1 mL emulsions, and the staining reaction was allowed to proceed for 1 h with gentle stirring at 25 °C. All representative images were obtained at a magnification of 20× using the stimulated emission depletion (STED) system.Scanning electron microscopy: The micromorphology of Pickering emulsion gels was observed with scanning electron microscopy (S4800, Hitachi, Tokyo, Japan) at 100 and 500× magnifications under the condition of high vacuum. The test samples were prepared by cutting the cross-sectional area from small parts of freeze-dried gels. Moreover, the internal microstructure of emulsion gels was characterized by a cryo-scanning electron microscope (FEI Quanta 450, Hiway, Beijing, China) under high vacuum conditions to in-depth explore the mechanism of the emulsion gel stabilization.

#### 2.4.3. Hardness

To evaluate the hardness of Pickering emulsion gel samples, we performed the penetration test with a texture analyzer (TMS-Pro 3000, FTC, Florida, FL, USA) and a flat bottom cylindrical probe (diameter 38 mm). Under the condition depicted in Section 2.3, Pickering emulsion gels were formed in a plastic sealed syringe with a height and diameter of 10 mm and 15 mm, respectively. The determination conditions were as follows: pre-test rate: 0.1 mm/s, test rate: 0.1 mm/s, post-test rate: 1 mm/s, compression distance: 5 mm. The hardness of the gel samples was defined as the maximum force which can penetrate to 50% height of the gel sample. Each sample was measured three times.

#### 2.4.4. Water Holding Capacity (WHC)

A centrifuge tube (50 mL) contained Pickering emulsion gels (25 mL) was used to determine their WHC. Pickering emulsion gel samples were centrifuged (Allegra X-30R, Beckman Coulter, Brea, CA, USA) at 8000× *g* for 15 min at 4 °C. The tea seed oil exposed to the surface of Pickering emulsion gel needs to be removed with filter paper. Hereafter, the released water above the gel samples after centrifugation was eliminated as described above, and WHC was calculated as follows:(1)$$\mathrm{WHC}(\%)=\frac{M_{C}}{M_{O}}\;\times \;100$$
where M_O_ is the gram of the gels before centrifugation and M_C_ is the gram of the gels which released water after centrifugation.

### 2.5. Statistical Analysis

The measurements were repeated three times in each experiment, and the data results were expressed as mean ± standard deviation, significant differences between samples were evaluated via Duncan’s test at a significance level of *p* < 0.05 by SPSS software (version 26, IBM software, Armonk, NY, USA).

## 3. Results and Discussion

### 3.1. Microstructure Observation of Pickering Emulsion Gel

Figure 1 shows the microstructures of Pickering emulsion gels under different conditions. It showed that droplet size decreased with increasing particle concentration, while the addition of kC made the droplet size significantly smaller. It is probably because kC formed a solid-like three-dimensional network in the continuous phase that prevented the emulsion droplets from coalescing and flocculation [25]. It is also possible that the addition of polysaccharides increased the viscosity of the continuous phase [26]. CLSM images of emulsion gels are presented in Figure 2. It can be seen from the green image that the oil droplet size of Pickering emulsion gel varied with kC concentration and particle concentration in accordance with Figure 1. With the addition of kC, the oil droplets dispersed more evenly, which indicates that the unabsorbed particles and kC in the continuous phase can effectively separate the oil droplets and form a more stable emulsion gel. This is similar to the results of Wang et al. In their study, W/O emulsions containing glucono-δ-lactone-induced casein gels in the inner phase showed higher resistance to destabilization caused by coalescence and sedimentation, than the equivalent non-gelled W/O emulsions [17]. It demonstrates that Pickering emulsion gels have an advantage in improving product quality compared to Pickering emulsions.

As shown in Figure 3a, the porous network structure of Pickering emulsion gel sample with 20% oil content could be observed by SEM after freeze-drying. Residual oil exists in the gel network due to the inability to remove tea oil during lyophilization, which is shown in the black shadow [27]. Hence, clearer images of gel structure can be obtained by using ether to remove residual oil droplets (Figure 3b). The change trend of Pickering emulsion gel before and after oil removal was similar. All emulsion gels showed a more obvious porous structure. When the concentration of kC was low, it emerged as a rough and loose structure. With the increase in kC concentration, the network structure was smoother and more compact, and the cavities became larger. The effect of particle concentration on network structure was evident at low kC concentrations and became smaller as the kC concentration increased. Addition of polysaccharides may promote the tight arrangement of droplets and increases with increasing polysaccharide concentration. On the one hand, the three-dimensional network formed by KC and particles becomes stronger with the increase in concentration, which hinders the aggregation of droplets. On the other hand, the addition of polysaccharides increases the viscosity of the continuous phase and makes the droplets more tightly arranged [28].

In order to further understand the formation mechanism of kC-based Pickering emulsion gel, Cryo-SEM was used to characterize the microstructure of emulsion gel. The results are shown in Figure 4. At all XG/Ly NPs concentrations, the network structure in emulsion gels with 2% kC concentration is more complex. Their droplets size was smaller than that of emulsion gels with 1% kC concentration, which was consistent with the previous results. A higher concentration of kC made the oil droplets separate into smaller sizes in the continuous phase, and an obvious network structure between the droplets and fine XG/Ly NPs uniformly attached to the surface of the droplets is observed in Figure 5b. The increase in kC concentration presented a tighter structure with the formation of a stronger interfacial network, which may be attributed to the higher absorption of Pickering emulsifiers across the oil-water interface. The colloidal particles around the oil droplets formed a dense interfacial layer, indicating that emulsion gel has the potential as a delivery system for bioactive to design better food formulations. It was noted that Pickering emulsion gel stabilized by octenylsuccinate quinoa starch granule with different gel networks by modulating the oil volume fraction could develop as a carrier for lutein. After 31 days of storage, the corresponding half-life times increased from 12 to 41 days [29].

### 3.2. Rheological Analysis of Colloidal Sol

To further study the gelation process of emulsion gel based on self-gelation of kC, a temperature scanning test was carried out. As shown in Figure 5a, when the concentration of XG/Ly NPs is 0.5%, the G′ of the emulsion with 0.5% kC is lower than G″, demonstrating that the Pickering emulsion did not form emulsion gel under this condition. While the gelation temperatures (T_gel_) of the emulsion with 1% and 2% kC are 23.41 °C and 33.51 °C, respectively, and the melting temperatures (T_melt_) are 35.91 °C and 51.68 °C, respectively (Table 1). This indicated that a higher kC concentration can make the emulsion form emulsion gel at a higher temperature during the cooling process and gave emulsion gel stronger heat resistance during the heating process.

The formation of emulsion gel was dramatically promoted when the particle concentration increased to a certain extent. When the concentration of kC was 1%, the T_gel_ and T_melt_ of emulsion gels increased to a different extent with the increase of XG/Ly NPs concentration. Similar results were observed in samples with kC concentration of 2%. However, compared with XG/Ly NPs, kC exerted greater effects on the T_melt_ of Pickering emulsion gel, indicating that kC played a leading role in the heat resistance of emulsion gel. In addition, the intersection between G′ and G″ was not observed in emulsion gel at the concentration of 2% kC and 2% XG/Ly NPs during the heating process (20–95 °C). Both the G’ and G” showed a decreasing trend with the increase in temperature, suggesting that Pickering emulsion gel was thermal irreversible under this condition [30]. According to the data in Table 1, thermal hysteresis became more obvious with the increase in kC concentration. The phenomenon maybe resulted from the multiphase Pickering emulsion system and different heat-mass transfer effect. Meanwhile, kC enhanced the thermal stability of Pickering emulsion gel and the capacity depended on XG/Ly NPs concentration.

Figure 6 shows the variation of storage modulus (G′) and loss modulus (G″) of emulsion gels with angular frequency (ω) at different concentrations of kC and XG/Ly NPs. It can be seen that the G′ and G″ of emulsion gels with 0.5–1% kC increased with the increase in frequency, and the trend slowed down with the increase in particle concentration. The G′ of emulsion with 0.5% kC is always lower than G″, indicating that there was a liquid-like structure and did not form any emulsion gel in the emulsions, which is consistent with Figure 5. When the concentration of kC further increased to 1% and 2%, the G′ of all emulsion gels is higher than G″ in a wide range of frequencies. In particular, the G′ and G″ of emulsion gels with 2% kC basically remain constant in a wide range of frequencies and are hardly affected by frequency, which indicated that Pickering emulsion gel system with a strong gel structure was formed [31]. However, Pickering emulsion with particle concentrations greater than 0.5% (Figure 6b,c) showed the crossing of G′ and G″ at higher frequency when the concentration of kC was 1%, which may result from the collapse of the solid structure into a liquid-like structure at a higher frequency.

In order to calculate the frequency dependence of G′ for the emulsion or emulsion gel, the power law model G′ = kω^n^ was used, and the results are shown in Table 2. The “*k*” value is related to the strength (elastic structure) of the gel network [32]. It can be seen from the table that the values of k for emulsion gel prepared with 2% kC were significantly larger than that of emulsion gel with 0.5–1% kC. The “*n*” value is a measure of the frequency dependence of the storage modulus (G′). The “*n*” value close to 0 is the characteristic of the gel containing more covalent cross-linking bonds [33]. The values of n for all emulsion gels are greater than zero (0.19–0.87), which indicated the formation of emulsion gels with non-covalent “physical” cross-linking bonds [34]. In addition, it can be seen from Figure 6d that the values of tan δ (G″/G′) of the emulsions with 0.5% kC increased and the absolute value is greater than 1 along with the increase in frequency, which implied that Pickering emulsions were not gelatinized. However, with the further increase of kC (1–2%), the tan δ almost does not change and the absolute values of tan δ are less than 1 with the increase in frequency. In addition, the absolute values of tan δ for emulsion gels with 2% kC are higher than that of emulsion with 1% kC. The results show that emulsion gel could be successfully constructed by adding kC to Pickering emulsion stabilized by XG/Ly NPs, and the strength and elasticity of emulsion gel could be regulated by particle stabilizer and kC concentration.

(1)Time scanning

As shown in Figure 7, we have executed a time scanning program at three temperature gradients (40 °C, 30 °C, and 20 °C). The G′ of the emulsion containing 0.5% kC is always lower than G″, which is consistent with the results of the temperature scan (Figure 6) and appearance (Figure 8), indicating that no emulsion gel was formed. When XG/Ly NPs concentration is 0.5%, the G’ of the emulsion gel containing 1% kC is always higher than G” at 20 °C, but no emulsion gel was formed at a temperature greater than 20 °C. The G′ of emulsion gel with 2% kC is always higher than G″ at 20 °C and 30 °C, but lower than G″ at 40 °C. The results confirmed the conclusion that the increase in kC concentration enhanced the heat resistance of emulsion gel during the temperature scanning. When the concentration of XG/Ly NPs is 1%, the G′ of emulsion gel with 1% kC is always higher than G″ at 20 °C and 30 °C, but lower than G″ at 40 °C, while the G′ of emulsion gel with 2% kC crossed with G″ at about 753 s (30 °C), indicating that gel melting occurred when emulsion gel was incubated at 30 °C. When the concentration of XG/Ly NPs is 2%, the G′ is basically coincident with G″. Currently, emulsion gels with higher particle concentration endowed emulsion gels with semi-solid structure, and the G′ of emulsion gel with 1% kC is always lower than G″ at 40 °C, but higher than G″ at 20 °C. The sample with 1% XG/Ly NPs differed in that G’ is crossed with G″ at about 1622s (30 °C).

According to the survey that GDL-induced emulsion gel can gelate at the incubation temperature of 40–60 °C, it can be observed that the G′ of emulsion gels basically coincided with G″ at about 112s (40 °C), crossed with G″ at about 948s (50 °C) and 1110s (60 °C), respectively [10]. In the present work, the G′ of Pickering emulsion gel with 2% kC almost over-lapped with G″ at 20 °C, demonstrating that higher concentration of XG/Ly NPs and kC gave emulsion gels more stable solid-like behavior. The G′ of emulsion gel is always lower than G″ at 40 °C and crossed with G″ at about 976s (30 °C). Compared with the results in Figure 7b, the melting time of Pickering emulsion gel is delayed, which indicated that the increase in particle concentration improved the stability of emulsion gel. To sum up, the gelation process of Pickering emulsion gel is a time- and temperature-dependent process, in which the dependence on temperature is more obvious.

### 3.3. Hardness

In Figure 8, the effect of kC and XG/Ly NPs concentration on the hardness of emulsion gels are shown. Obviously, the increase in kC concentration contributes to the formation of emulsion gel and significantly enhanced the strength of emulsion gel. However, when the concentration of kC is 0.5%, the emulsions did not form gels as the concentration of XG/Ly NPs is low (0.5–1%). The appearance of the fluidity emulsion can also be seen in the illustration in Figure 8. When the concentration of XG/Ly NPs increased to 2%, a weak emulsion gel was formed. At this time, the gel structure of the emulsion was mainly dominated by XG/Ly NPs. When the concentration of kC increased to 1%, emulsion gels were formed at all XG/Ly NPs concentrations with lower hardness. When kC further increased to 2%, the hardness of Pickering emulsion gels formed by different XG/Ly NPs concentrations was obviously different. Especially when the XG/Ly NPs concentration is 2%, the hardness of emulsion gel reached the highest (2.9 N). It can be attributed to the formation of a strong interfacial film by the high concentration of XG/Ly NPs, and the elastic modulus of emulsion gel was closer to solid when 2% kC was added. In addition, smaller oil droplet also helps to improve the strength of Pickering emulsion gel [35]. The results showed that emulsion gels with various hardness can be formed by adjusting the concentration of kC and XG/Ly NPs, which has important guiding significance for designing foods with different hardness to meet consumer groups of different ages.

### 3.4. Water-Holding Capacity (WHC)

In order to gain more understanding of the structure of Pickering emulsion gel, WHC is used as an indicator to evaluate the capability of the gel matrix to fix the water. Figure 9 shows the WHC of emulsion gels under different concentrations of kC and XG/Ly NPs. Except for emulsions that have not formed a gel (kC 0.5%), the WHC of all Pickering emulsion gels is over 97%. It is not difficult to find that the WHC of the emulsion with a 2% kC concentration (about 99% on average) is higher than that of the Pickering emulsion with a 1% kC concentration (about 98% on average), indicating that a tighter network structure was formed inside Pickering emulsion gel with the increase in kC concentration. The tight network prevents the migration of water in emulsion gel. However, there is basically no difference in WHC of Pickering emulsion gels with 1 and 2% kC because of their high WHC capacity. As it can be seen from Pickering emulsion gels with 1% kC, the WHC presented a gradually increasing trend with the increase in particle concentration. This phenomenon was probably attributed to more XG/Ly NPs competing with kC for moisture so that more moisture was retained in emulsion gel [36]. Moreover, the increase of XG/Ly NPs may improve the viscosity of emulsion gel, which delayed the flow rate of water, and thus reduced the precipitation of water. At this time, kC may play a dominant role in the structure of the continuous phase. In addition, the higher WHC of Pickering emulsion gel also indicated that its internal structure can retain more nutrients, which provided a theoretical basis for utilizing tea oil during processing [37].

## 4. Conclusions

In this work, Pickering emulsion system was regulated by kC self-gelation. The droplet size of emulsion gel gradually decreased with the increase of kC and XG/Ly NPs concentration. Microscopic results showed that the porous network structure formed in the continuous phase of Pickering emulsion gel could effectively separate the oil droplets from each other, serving to maintain the stability of Pickering emulsion gel. The gelation process of emulsion gel is a time- and temperature-dependent manner. The hardness of Pickering emulsion gel is up to 2.9 N, which could be adjusted by the concentration of kC and XG/Ly NPs. The results indicated that kC self-gelling can construct and control the structure and characteristics of emulsion gels, thus providing a theoretical basis for various practical applications of functional Pickering emulsion gels.

## Figures and Tables

**Figure 1 foods-11-03757-f001:**
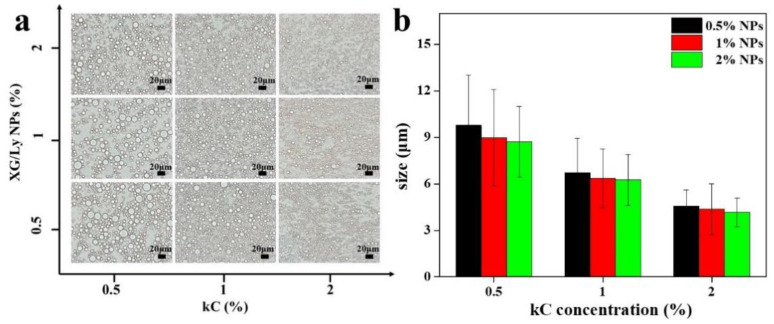
Optical microscope (**a**) and size distribution (**b**) of Pickering emulsion gel stabilized by XG/Ly NPs with kC.

**Figure 2 foods-11-03757-f002:**
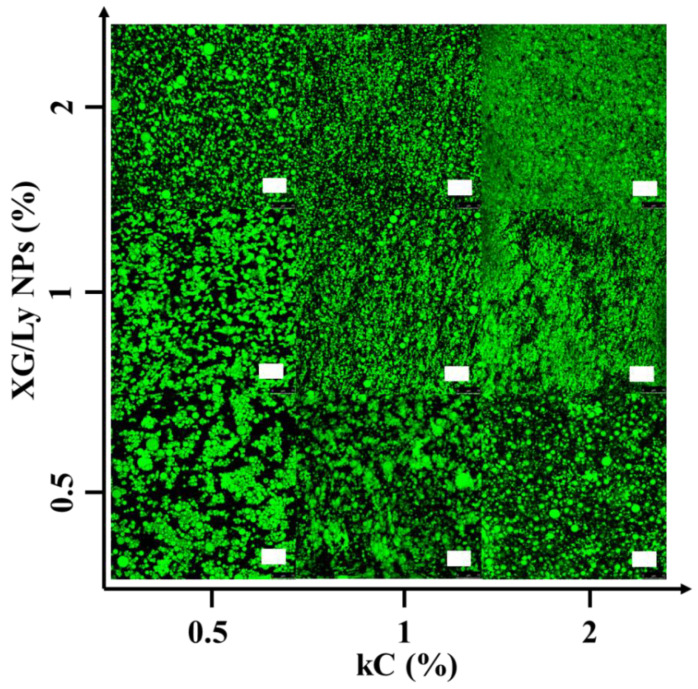
CLSM images of Pickering emulsion gel stained by Nile red prepared, scale bar 75 μm.

**Figure 3 foods-11-03757-f003:**
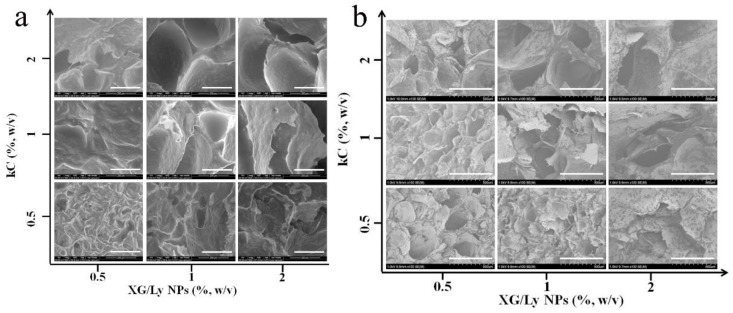
SEM images of Pickering emulsion gels with different concentrations of kC and XG/Ly NPs. (**a**), oil droplets reserve, scale bar 200 μm; (**b**), oil droplets remove, scale bar 500 μm.

**Figure 4 foods-11-03757-f004:**
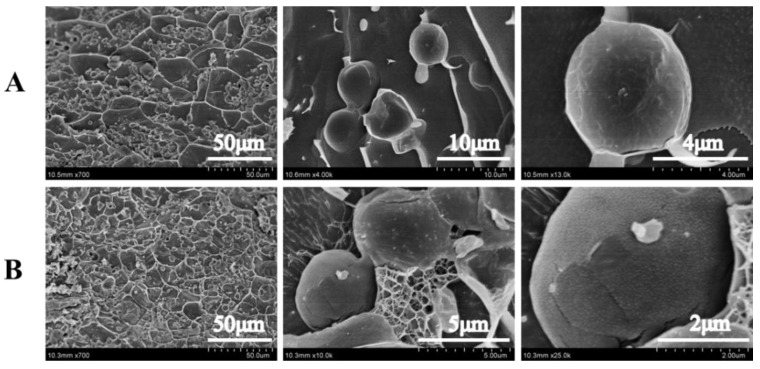
Cryo-SEM images of Pickering emulsion gels with different concentrations of kC ((**A**) 1%, (**B**) 2%) at 1% XG/Ly NPs.

**Figure 5 foods-11-03757-f005:**
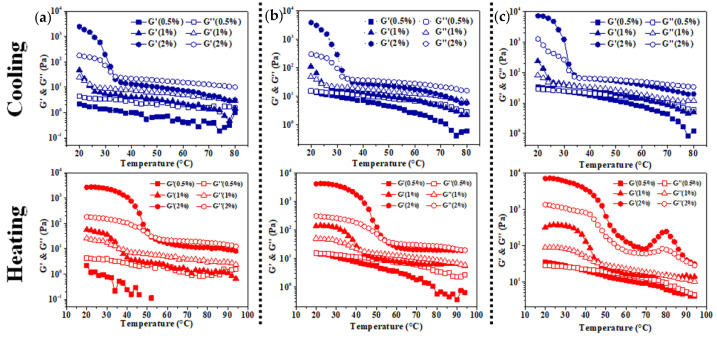
Gelation process and gel-melting process of Pickering emulsion gels stabilized by XG/Ly NPs ((**a**) 0.5%, (**b**) 1%, (**c**) 2%) with kC (0.5%, 1%, 2%).

**Figure 6 foods-11-03757-f006:**
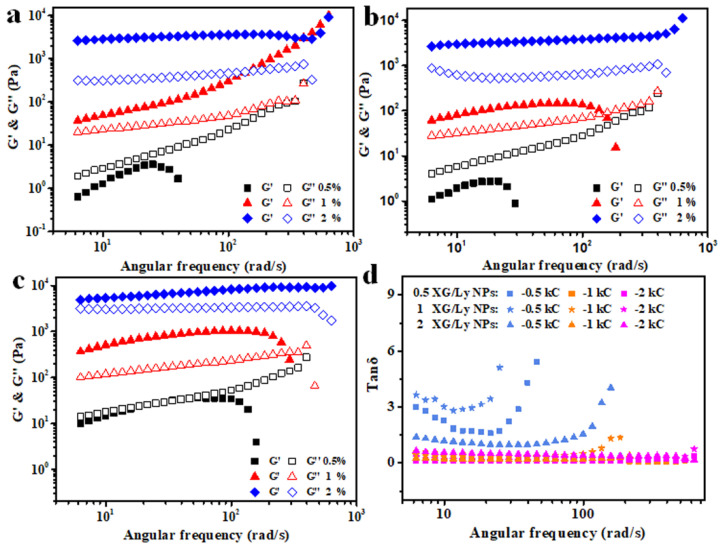
Frequency dependence of G′ and G″ for Pickering emulsion gels prepared by different concentration of XG/Ly NPs (**a**-0.5%, **b**-1%, **c**-2%) at various concentrations of kC as a function of angular frequency and the frequency dependence of tan δ (**d**) for emulsion gels.

**Figure 7 foods-11-03757-f007:**
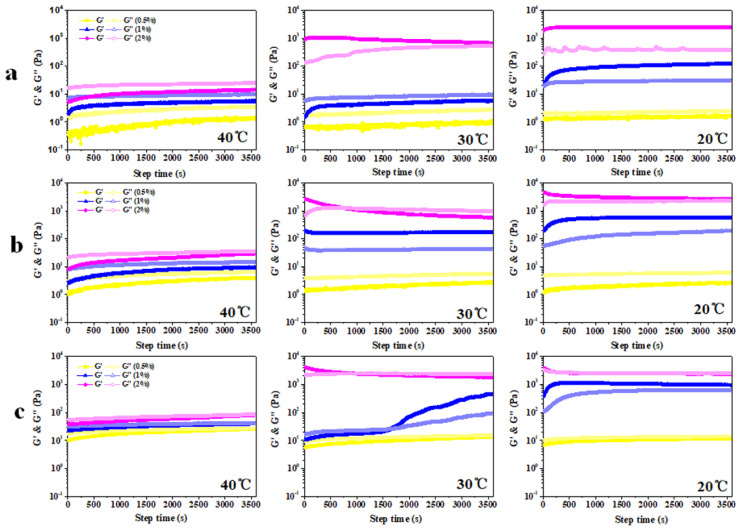
G′ and G″ of Pickering emulsion gel prepared by different concentrations of XG/Ly NPs ((**a**) 0.5%, (**b**) 1%, and (**c**) 2%) at various concentrations of kC under different temperature conditions.

**Figure 8 foods-11-03757-f008:**
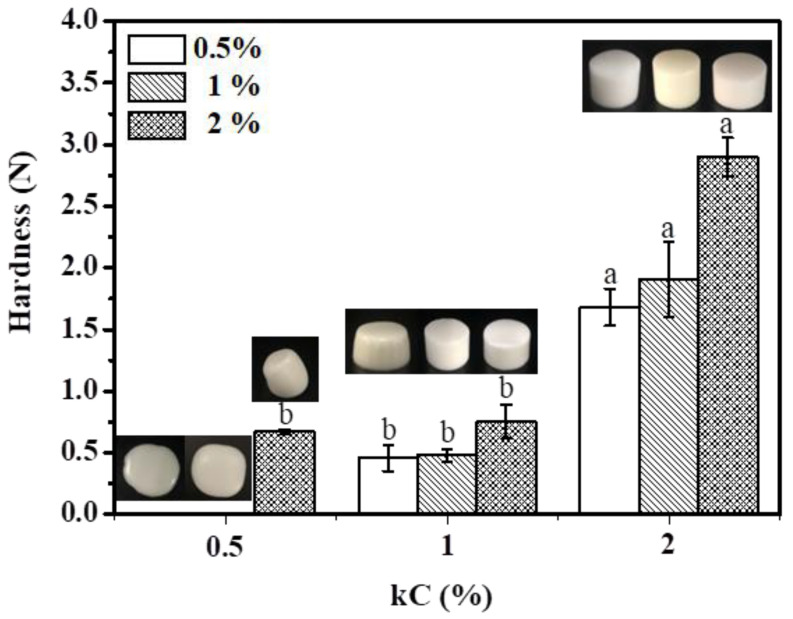
The gel strength and appearance of Pickering emulsion gels. Different letters represent significant differences (*p* ≤ 0.05).

**Figure 9 foods-11-03757-f009:**
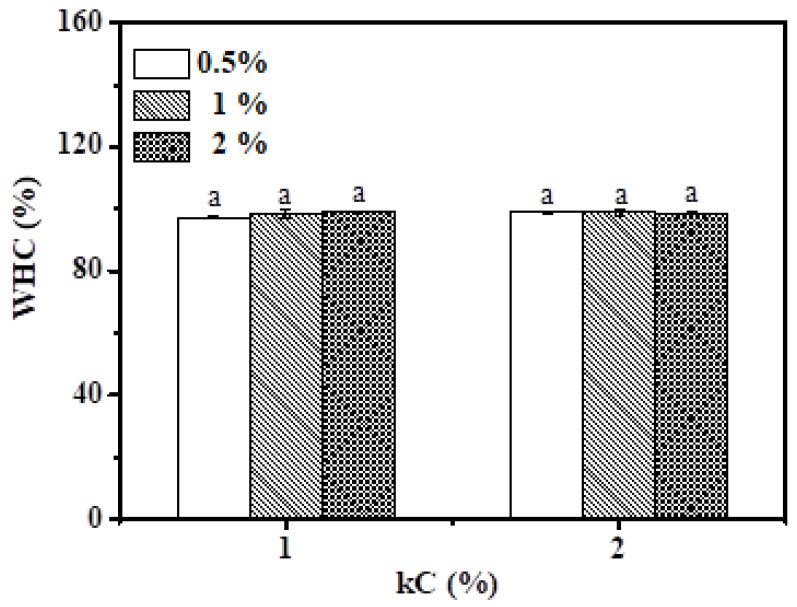
The water-holding capacity of Pickering emulsion gels. Different letters represent significant differences (*p* ≤ 0.05).

**Table 1 foods-11-03757-t001:** Gelation (T_gel_) and gel-melting (T_melt_) temperature (°C) of Pickering emulsion gels in different XG/Ly NPs concentrations coupled with different concentrations of kC.

XG/Ly NPs (%)	kC (%)					
	0.5		1		2	
	T_gel_	T_melt_	T_gel_	T_melt_	T_gel_	T_melt_
**0.5**	—	—	23.41	35.91	33.51	51.68
**1**	—	20.83	25.65	42.20	34.07	54.94
**2**	29.26	33.22	36.81	47.13	37.83	—

**Table 2 foods-11-03757-t002:** Power law modelling parameters of frequency dependent behavior for emulsion gels.

XG/Ly NPs (%)		kC (%)				
	k			n		
	0.5	1	2	0.5	1	2
**0.5**	0.43	1.98	688.62	0.87	0.81	0.41
**1**	4.18	7.22	633.56	0.19	0.57	0.44
**2**	4.46	36.94	1680.92	0.57	0.62	0.44

## Data Availability

The data presented in this study are available on request from the corresponding author.

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
