# Peer review of "Carrageenan-Based Pickering Emulsion Gels Stabilized by Xanthan Gum/Lysozyme Nanoparticle: Microstructure, Rheological, and Texture Perspective"

_foods, 2022, doi:10.3390/foods11233757_

Round 1

Reviewer 1 Report

The topic is very interesting; however, the article needs to be further improved and some clarification should be done.

The introduction needs to be improved and fully checked for English, especially with regards to the verb tenses that are used. Mainly the intro talks about “emulsion gels” and not “Pickering emulsion gel”, therefore, I suggest authors give a bit more background on the differences/advantages of Pickering emulsion gel. In fact, throughout the manuscript authors talk about the system being a Pickering gel but has not provided any info/prove to assure the system is indeed a Pickering emulsion. Authors should also stick to the use of a single terminology, sometimes they use “fluid Pickering” and at other times “liquid Pickering”.

For the temperature sweep measurements why did authors start the analysis with the “liquid” at 80 C? Why not directly evaluate the gel? If the idea was to see the gelation should the analysis with the “liquid” have started at ambient temperature instead of 80C? Also, how was the strain of 1% chosen? Randomly? The Water holding capacity (WHC) analysis needs to be better explained as it is confusing.

I suggest author used a software to calculate the average droplet size as well as the size distribution from the micrographs so that the author can discuss that section “quantitatively” instead of “qualitatively”.

Other commets:

Line 14:” effect” on

Line 17: which property?

Line 37: cosmetics fields? Why would “food-grade” emulsions be of interest to the cosmetic industry?

Line 39: Saving cost? Doesn’t that depend directly on the material used rather than the system itself?

Line 43: were formed, define LPs

Line 81: State everything or nothing and “so on” is not appropriate

Line 87:XG?

Line 90: Tea oil? Which tea?

Line 165: What is meant by “The Pickering emulsion gels within a centrifuge tube (50 mL volume) were fabricated to determine their WHC”?? Rephrase as it does not make sense

Line 212: Change law?

Line 213: Verb tense

Figure 3 and 4: I suggest merging these as a and b to facilitate the comparisons

Line 300:  Discuss the results in terms of rad/s (graphs axis) instead f in Hz

Author Response

Dear,

Thank you for your useful comments and suggestions on our manuscript. We have modified the manuscript accordingly and the detailed corrections are listed below point by point.  

  1. The introduction needs to be improved and fully checked for English, especially with regards to the verb tenses that are used. Mainly the intro talks about “emulsion gels” and not “Pickering emulsion gel”, therefore, I suggest authors give a bit more background on the differences/advantages of Pickering emulsion gel. In fact, throughout the manuscript authors talk about the system being a Pickering gel but has not provided any info/prove to assure the system is indeed a Pickering emulsion. Authors should also stick to the use of a single terminology, sometimes they use “fluid Pickering” and at other times “liquid Pickering”.

Reply: Thank you for your valuable advice. We have revised the introduction accordingly and carefully checked the language. Besides, we have emphatically introduced more background on the differences/advantages of Pickering emulsion gel. The introduction part has been organization according to your advices. Additionally, the study of Pickering emulsion gel is system further research of Pickering emulsion stabilized by xanthan gum/lysozyme nanoparticle, in which we have published many papers [1-5]. Some other confused express in the paper have been revised.

References

[1]Wei Xu, Weiping Jin, Kunling Huang, Lu Huang, Yucui, Juang Li, Xinfang Liu, Bin Li. Interfacial and emulsion stabilized behavior of lysozyme/xanthan gum nanoparticles. International Journal of Biological Macromolecules, 2018, 117, 280-286.

[2] Zhifan Li, Shuqing Zheng, Cong Zhao, Mengru Liu, Zirui Zhang, Wei Xu, Denglin Luo, Bakht Ramin Shah. Stability, microstructural and rheological properties of Pickering emulsion stabilized by xanthan gum/Lysozyme nanoparticles coupled with xanthan gum, International Journal of Biological Macromolecules, 2020, 165, 2387-2394.

[3]Wei Xu, Zhifan Li, Haomin Sun, Shuqing Zheng, He Li, Denglin Luo, Yingying Li, Mengyuan Wang, Yuntao Wang,High Internal-phase Pickering Emulsions Stabilized by Xanthan Gum/Lysozyme Nanoparticles: Rheological and Microstructural Perspective, Frontiers in Nutrition,2022, 8, 744234.

[4]Wei Xu, Zhifan Li, He, Li, Haomin Sun, Shuqing Zheng, Denglin Luo, Yingying Li, Yuntao Wang, Bakht Ramin Shah, Stabilization and microstructural network of Pickering emulsion using different xanthan gum/lysozyme nanoparticle concentrations, LWT, 2022, 160, 113298.

[5]Wei Xu, Haomin Sun, He, Li, Zhifan Li, Shuqing Zheng, Denglin Luo, Yuli Ning, Yuntao Wang, Bakht Ramin Shah, Preparation and characterization of tea oil powder with high water solubility using Pickering emulsion template and vacuum freeze-drying, LWT-Food Science and Technology, 2022, 160, 113330.

  1. For the temperature sweep measurements why did authors start the analysis with the “liquid” at 80 C? Why not directly evaluate the gel? If the idea was to see the gelation should the analysis with the “liquid” have started at ambient temperature instead of 80C? Also, how was the strain of 1% chosen? Randomly? The Water holding capacity (WHC) analysis needs to be better explained as it is confusing.

Reply: Thank you for your valuable advice. The temperature sweep measurements in situ simulated the preparation condition of Pickering emulsion gel which further explain the gelation process. It is knowledge that CRG was dissolve at 70oC and gelatinize when cooled to room temperature. For better to dissolve and be gel, we used the temperature of 80oC. The reason of choosing the strain of 1% is that to all measurements with the strain were performed within the linear viscoelastic region. We have repeated the water holding capacity (WHC) tests which displayed in the paper.

  1. Line 14: “effect” on

Reply: Thank you for your valuable advice. The sentence has been revised.

Line 17: which property?

Reply: The sentence has been revised as ‘rheological properties’.

Line 37: cosmetics fields? Why would “food-grade” emulsions be of interest to the cosmetic industry?

Reply: Because of many lotion, cream and other cosmetics are kind of emulsion systems.

Line 39: Saving cost? Doesn’t that depend directly on the material used rather than the system itself?

Reply: Thank you for your valuable suggestion, we have revised the sentence and removed the ‘saving cost’.

Line 43: were formed, define LPs

Reply: Thank you for your valuable suggestion, we have not used abbreviation for XG/Ly NPs before in the published paper.

Line 81: State everything or nothing and “so on” is not appropriate

Reply: We have revised the sentence according to your kind suggestion.

Line 87:XG?

Reply: We have revised the sentence as ‘xanthan gum (XG)’.

Line 90: Tea oil? Which tea?

Reply: We have checked the material and revised the tea seed oil information. Tea seed oil (saturated fatty acid 10%, monounsaturated fatty acid 80%, and polyunsaturated fatty acids 10%) was purchased from was obtained in a local supermarket which produced from Luda Green organic camellia oil Co. Ltd.

Line 165: What is meant by “The Pickering emulsion gels within a centrifuge tube (50 mL volume) were fabricated to determine their WHC”?? Rephrase as it does not make sense

Reply: Thank you for your valuable suggestion, we have revised the sentence as ‘A centrifuge tube (50 mL volume) contained Pickering emulsion gels were used to determine their WHC’.

Line 212: Change law?

Reply: Thank you for your valuable suggestion, the mistake has been revised.

Line 213: Verb tense

Reply: The sentence has been revised.

Figure 3 and 4: I suggest merging these as a and b to facilitate the comparisons

Reply: We have merged Figure 3 and 4. The other figures has also been organized.

Line 300:  Discuss the results in terms of rad/s (graphs axis) instead f in Hz

Reply: Thank for your kind suggestion. Commonly, both of angular frequency (rad/s) and frequency (Hz) are often used in the frequency sweep measurement. The variation of storage modulus (G′) and loss modulus (G″) with angular frequency (ω) or frequency illustrate the viscoelastic properties as suffer shear force. The formula of angular frequency are listed below:

ω=2π/T=2πf.

Where ω: angular frequency;

      T: period;

      f: frequency.

Reviewer 2 Report

The articles detail the formation of emulsion gels using three levels of xanthan gum/lysozyme nanoparticles and kappa Carrageenan. While it is clear that the addition of the carrageenan modifies the morphological and rheological behavior of the obtained emulsions, as observed by the morphological (optical microscopy, confocal fluorescence microscopy, SEM, CryoSEM), rheological, mechanical and water retention capabilities of the obtained gels.

While this work has an interesting scientific merit, there are different issues that must be solved in the paper before it is acceptable for publication

In general, English shall be thoroughly revised, there are several syntactic and grammatical mistakes and awkward phrasing. For example, L36. 

Introduction 

The importance of the work shall be improved, as it is not clear the intended use, besides edible emulsions. If there is a potential market, please indicate some data supporting your point, otherwise a more fundamental science approach is valid as well.  

It is unclear the extent or nature of the emulsion destabilization and the mechanisms proposed for its stabilization shall be detailed, with their respective references. 

The difference between the proposed and previous uses of k-Carrageenan shall be mentioned and explained as well, as it is not clear. The variables of interest can be mentioned at the end of the section, right before detailing the techniques, as it helps to better assess and understand the assumptions of this work. The use of stability as a property to improve is not coherent with the rest of the work, stability tests are not mentioned in the following sections, therefore results would not be conclusive for this property.

L43. Even though this description was first mentioned in the abstract, please include the meaning of XG/Ly NPs in this line 

L46. Please indicate the time it is stable, what kind of stability is compromised, is it chemical or physical or both. If it is only physical, please reconsider the example given in L48.  

L46-L47. It is indicated that different approaches are used to improve the emulsion stability, so far only one approach, emulsion gels, is mentioned. Please include at least another approach and its reference. 

L51. Please include a reference for the definition of emulsion gel. 

L56-59. Please be more specific in the improvements provided by these gels, which was the improvement in stability, was it verified by a limited increase in droplet size or by qualitative evaluation of droplet morphology and a lack of phase separation, it is not required to do a complete briefing of the articles, but please indicate some basic data to support your point. 

L61. Please indicate a couple of examples and their reference. 

L63. Please provide a reference to the resistance of digestive enzymes. 

L74. Please indicate if it has been used in emulsions before, and in which type of emulsions (oil type or content) and their improvement. 

L76-84. Please stress the main objective of this work and the difference between this use of k-Carrageenan and previous uses of this polysaccharide. 

L79. What is understood by “good”. Please use a more neutral term. 

L81. Please be more descriptive, which techniques are encompassed by the sentence “so on” 

Materials and methods 

The methods are described, however, considering that the use of kC was proposed to improve gel stability it is remarkable that no stability test is described in this section, while the use of rheological measurements can be associated with the kinetic stability of the emulsion, a direct measurement of this property would provide a more robust assessment. 

L97. Please indicate the compound used to adjust the pH. 

L98. The solution was dialyzed against deionized water, right? If so, please indicate it in the manuscript. 

L103. There are two components XG/Ly NPs (using a 1:1 ratio, according to L 96) and kC, therefore I do not understand the final concentrations mentioned in this line. 

L130. An amplitude sweep will be used to corroborate that this deformation is in the LVR?  

L138-147. Which components are dyed by Nile red, and which are dyed by Nile blue? Please specify. 

L153. Please be a little bit more specific in this section. How were the samples loaded to the chamber and sublimated. 

L166-169, According to this description, the emulsion is not stable to the high shear exerted in this test, just for curiosity did the authors evaluated stability to centrifugation of the emulsion? Did the authors quantified the amount of tea oil that coalesced? Did this weight have an impact on the measured value? 

Results and discussions 

It is important to specify if the percentages are in a weight base, while it is highly likely, this specification helps to improve the understanding of this work. 

This section requires a more in-depth discussion of the results, the phenomena are mentioned shallowly, they need to be further discussed, with the proper references. While the comparison with previous works needs to broaden the literature research.  

L186. Please provide a reference 

L190. Is it kC or KC, please be consistent. 

L190. Which compound is colored green? Or is it the compound color of Nile red and Nile blue? Please specify. 

L194. I think “Wang” is written with a capital W. Also, include the number of the reference, rather than its year. 

L191-192. Please be more specific in the description of Wang et al. It is unclear what did the original authors developed, or which is similar or different to the current work. 

L204-205. Please indicate which component is colored green. Considering the potential of this technique, were the authors able to identify the location of the XG/Ly NPs in relation to the oil interface.  

L210. What do authors associate with a dark shadow, it is unclear which part of Fig. 3 they are referring to. 

L212. What is understood as the “change law. Please use a clearer term.  

L213. What do the authors mean as more obvious network structure, another sentence would be preferred in this case. 

L220-L222. Please provide references to these observations. 

L225. Using arrows to point at the tea oil droplets help to better understand the information conveyed. 

L234. Please reference the previous result. 

L240. Please provide a reference to the affirmation of the adsorption towards the interface. It is hard to identify the different structures in the image, could you please indicate which are the XG/Ly NPs in Fig. 5A-B. 

L230-242. The discussion can be more profound, please include a more in-depth information about the phenomena observed and the reasons that could lead to these structures to be used in food formulations. 

L244. Please provide visual helps to better understand the structures inside Fig. 5A.B 

L274. It appears as if G’ and G’’ of the sample with 2 % XG/Ly and 2% kC will cross after 95 °C. Rather than being irreversible, the gel does not melt in the temperature range studied. Otherwise, if there is an irreversibility; please explain the process in more detail and provide the corresponding references. 

L275. In which sense the result is coherent with the reference provided. Please indicate, briefly, how the results reported in that reference can help to explain or compare to the results reported in this work. 

L277-279. Please elaborate a little bit on the phenomenon present in this thermal process, provide references. 

L282. It is odd that G’ decreases at high frequencies, please verify that the system is effectively in the Lineal viscoelastic range, as this zone decreases at higher frequencies. Also tan (delta) should not be below 1, apparently the data has inertia correction problems, as G’ sometimes adopt negative values perhaps a larger geometry would avoid these issues.  

L299-302. Please provide a reference, it is also not clear why an increase in the content of XG/Ly NPs reduces G’ of the 1 % kC samples at high frequency. 

L302-320. While this comparison is interesting, please verify the limits of the Power Law equation, as it is used in the Lineal Viscoelastic Regime and this zone is not present for all the samples. Please revise and correct the analysis appropriately. It is uncommon to have negative values for tan(delta), please verify. 

L323-351. Results indicate that even at the highest level of kC , the sample can melt at longer times after 40 °C, however, this behavior is clearly affected by the presence of XG/Ly. However, it is unclear the underlying phenomena that is occurring in this case, please include more references, compare to previous works, and explain what is happening in a clearer way. 

L350-351. Please redact the line in a more technical writing style. 

L357. Please indicate why hardness is relevant to this work. 

L371-373. The formation of a 3 D network of kC, which is mentioned in previous paragraphs shall not be neglected to explain hardness. Also please indicate if a finer oil droplet would affect this property. Provide references. 

L387. Please supply a clearer explanation, although according to the standard deviations there is no statistically significant difference between 1 and 2% kC samples. 

L397. Please supply a reference. 

L399. Please supply a reference. 

L401. It would be good to do a very brief summary of the importance of these results as a way to close the results and previous to conclusions. 

Conclusions 

This section is brief and indicates the most important results of the work, however there is a disconnect between this part and the introduction, as emulsion stability is important in the latter, while it is clearly absent in the rest of the work. At least the introduction shall be reformulated to address this problem. 

L417. The WHC value of the gels do not differ too much, but it would be good to see the actual values and their standard deviations to be sure. 

Author Response

Dear,

Thank you for your useful comments and suggestions on our manuscript. We have modified the manuscript accordingly and the detailed corrections are listed below point by point.

The articles detail the formation of emulsion gels using three levels of xanthan gum/lysozyme nanoparticles and kappa Carrageenan. While it is clear that the addition of the carrageenan modifies the morphological and rheological behavior of the obtained emulsions, as observed by the morphological (optical microscopy, confocal fluorescence microscopy, SEM, CryoSEM), rheological, mechanical and water retention capabilities of the obtained gels. While this work has an interesting scientific merit, there are different issues that must be solved in the paper before it is acceptable for publication

In general, English shall be thoroughly revised, there are several syntactic and grammatical mistakes and awkward phrasing. For example, L36. 

Reply: Thank you for your valuable advice. We have revised the paper and carefully checked the language. Besides, we have also ask for an international friend who is native English speaker to help us revise the language. The sentence of L36 has also been revised.

Introduction 

The importance of the work shall be improved, as it is not clear the intended use, besides edible emulsions. If there is a potential market, please indicate some data supporting your point, otherwise a more fundamental science approach is valid as well.  

Reply: Thank you for your valuable advice. We have revised the introduction part and especially emphasized the practical application and scientific significance of the research.

It is unclear the extent or nature of the emulsion destabilization and the mechanisms proposed for its stabilization shall be detailed, with their respective references. 

Reply: We have rewrite the introduction part and reorganize the strucutre. We have added some information in the part, such as ‘Pickering emulsion gels combine the virtues of both Pickering emulsions and gels. Compare with Pickering emulsions, emulsion gels are stable against droplet coalescence, Ostwald ripening, and phase separation’.

The difference between the proposed and previous uses of k-Carrageenan shall be mentioned and explained as well, as it is not clear. The variables of interest can be mentioned at the end of the section, right before detailing the techniques, as it helps to better assess and understand the assumptions of this work. The use of stability as a property to improve is not coherent with the rest of the work, stability tests are not mentioned in the following sections, therefore results would not be conclusive for this property.

Reply: Thank you for your kind advice. First, we added the information of kC of Pickering emulsion. Besides, the comparative study were also strengthened in the following results and conclusive part.

L43. Even though this description was first mentioned in the abstract, please include the meaning of XG/Ly NPs in this line 

Reply: We have defined the the meaning of XG/Ly NPs according to your kind suggestion.

L46. Please indicate the time it is stable, what kind of stability is compromised, is it chemical or physical or both. If it is only physical, please reconsider the example given in L48.  

Reply: According to your kind suggestion, we have revised the sentence and added the physical reason for the stability of Pickering emulsions stabilized by XG/Ly NPs.

L46-L47. It is indicated that different approaches are used to improve the emulsion stability, so far only one approach, emulsion gels, is mentioned. Please include at least another approach and its reference. 

Reply: According to your kind suggestion, we have provide some approaches to improve the emulsion stability, such as polysaccharide intervention and increasing particle stabilizer concentration.

L51. Please include a reference for the definition of emulsion gel. 

Reply: We have added the definition of Pickering emulsion gel in the introduction part because we mainly describe Pickering emulsion gel. The detailed information is ‘Pickering emulsion gel is one phase dispersed in another enable the design of two-phase systems, where the microstructures and viscoelastic properties can be tailored’.

L56-59. Please be more specific in the improvements provided by these gels, which was the improvement in stability, was it verified by a limited increase in droplet size or by qualitative evaluation of droplet morphology and a lack of phase separation, it is not required to do a complete briefing of the articles, but please indicate some basic data to support your point. 

Reply: Thank you for your kind advice. The related introduction part has been rewritten.

L61. Please indicate a couple of examples and their reference. 

Reply: Thank you for your kind advice. The second paragraph of introduction part been rewritten.

L63. Please provide a reference to the resistance of digestive enzymes. 

Reply: We have provide a reference according to your kind advice.

L74. Please indicate if it has been used in emulsions before, and in which type of emulsions (oil type or content) and their improvement. 

Reply: The information of oil type is provided in the materials part.

L76-84. Please stress the main objective of this work and the difference between this use of k-Carrageenan and previous uses of this polysaccharide. 

Reply: According to your kind suggestion, we emphasize the main objective of this work in the last paragraph of introduction part. It noted that the results to broaden the application of polysaccharides represented by kC in the construction of new gel-related food structures transforming from sol to gel state.

L79. What is understood by “good”. Please use a more neutral term. 

Reply: The sentence has been revised according to your kind advice.

L81. Please be more descriptive, which techniques are encompassed by the sentence “so on” 

 Reply: The sentence has been revised according to your kind advice.

Materials and methods 

The methods are described, however, considering that the use of kC was proposed to improve gel stability it is remarkable that no stability test is described in this section, while the use of rheological measurements can be associated with the kinetic stability of the emulsion, a direct measurement of this property would provide a more robust assessment. 

Reply: Thank you for your kind advice. We I fully agree with your suggestions. As the Pickering emulsion has transformed from sol to gel state. The stability was greatly improved. That is why we do no carry the test.

L97. Please indicate the compound used to adjust the pH. 

Reply: We have revised the sentence as ‘XG and Ly mixture solutions with concentration 1.0 mg/mL respective were prepared and adjusted pH to 11.8’.

L98. The solution was dialyzed against deionized water, right? If so, please indicate it in the manuscript. 

Reply: According to your kind suggestion, we have revised the sentence.

L103. There are two components XG/Ly NPs (using a 1:1 ratio, according to L 96) and kC, therefore I do not understand the final concentrations mentioned in this line. 

Reply: We are sorry to describe the puzzled sentence. We have revised the sentence as ‘kC was dispersed in XG/Ly NPs solution and heated by stirring in a water bath at 70°C for 20 min until the kC was completely dissolved. It was stirred until it was cooled to room temperature’,

L130. An amplitude sweep will be used to corroborate that this deformation is in the LVR?  

Reply: Before Frequency sweep measurements, we have carried amplitude sweep to check the LVR. A fixed strain of 0.1% was used to guarantee within the range of LVR. Besides, our other similar studies have also displayed bellow to check.

References

[1] Zhifan Li, Shuqing Zheng, Cong Zhao, Mengru Liu, Zirui Zhang, Wei Xu, Denglin Luo, Bakht Ramin Shah. Stability, microstructural and rheological properties of Pickering emulsion stabilized by xanthan gum/Lysozyme nanoparticles coupled with xanthan gum, International Journal of Biological Macromolecules, 2020, 165, 2387-2394.

[2]Wei Xu, Zhifan Li, Haomin Sun, Shuqing Zheng, He Li, Denglin Luo, Yingying Li, Mengyuan Wang, Yuntao Wang,High Internal-phase Pickering Emulsions Stabilized by Xanthan Gum/Lysozyme Nanoparticles: Rheological and Microstructural Perspective, Frontiers in Nutrition,2022, 8, 744234.

[3]Wei Xu, Zhifan Li, He, Li, Haomin Sun, Shuqing Zheng, Denglin Luo, Yingying Li, Yuntao Wang, Bakht Ramin Shah, Stabilization and microstructural network of Pickering emulsion using different xanthan gum/lysozyme nanoparticle concentrations, LWT, 2022, 160, 113298.

L138-147. Which components are dyed by Nile red, and which are dyed by Nile blue? Please specify. 

Reply: Thank you for your kind advice. We have revised the methods. We commonly used Nile red and Nile blue to stain oil and XG/Ly NPs. For the paper, we just used Nile red to stain tea oil because it difficult to oil phase and t XG/Ly NPs phase in the Pickering emulsion gel.

L153. Please be a little bit more specific in this section. How were the samples loaded to the chamber and sublimated. 

Reply: Thank you for your kind advice. The corresponding information has been provided.

L166-169, According to this description, the emulsion is not stable to the high shear exerted in this test, just for curiosity did the authors evaluated stability to centrifugation of the emulsion? Did the authors quantified the amount of tea oil that coalesced? Did this weight have an impact on the measured value? 

 Reply: It noted that Pickering emulsion is commonly is not stable to the high shear exerted. But in this study, Pickering emulsion gel is formed based kC gelatinization. The solidification greatly improved the stability of Pickering emulsion. The Pickering emulsion gel could suffer high shear and centrifuge.

Results and discussions 

It is important to specify if the percentages are in a weight base, while it is highly likely, this specification helps to improve the understanding of this work. 

Reply: Thank you for your kind advice. We have check the percentages unit in the paper.

This section requires a more in-depth discussion of the results, the phenomena are mentioned shallowly, they need to be further discussed, with the proper references. While the comparison with previous works needs to broaden the literature research.  

Reply: Thank you for your kind advice. We have carefully revised the part. And try to further discuses the results in the paper. Besides, we also contrastive analysis with previous works to broaden the literature research.  

L186. Please provide a reference 

Reply: We added a reference in the appropriate place to support the information.

L190. Is it kC or KC, please be consistent. 

Reply: The mistake has been revised.

L190. Which compound is colored green? Or is it the compound color of Nile red and Nile blue? Please specify. 

Reply: The colored green is Nile red stained tea oil in the Pickering emulsion gel. The information has been provided in the paper.

L194. I think “Wang” is written with a capital W. Also, include the number of the reference, rather than its year. 

Reply: Thank you for your kind advice. The mistake has been revised.

L191-192. Please be more specific in the description of Wang et al. It is unclear what did the original authors developed, or which is similar or different to the current work. 

Reply: We specific in the description of Wang et al in the paper according to your kind suggestion.

L204-205. Please indicate which component is colored green. Considering the potential of this technique, were the authors able to identify the location of the XG/Ly NPs in relation to the oil interface.  

Reply: Thank you for your kind advice. The component was oil in the Pickering emulsion gel displayed colored green. We have used CLSM to locate XG/Ly NPs in relation to the oil interface before (Fig.1). While in the paper, it is difficult to identify the location of the XG/Ly NPs for high viscosity of kC in the gel system. Therefore, we just displayed partial results in the paper. Some results we displayed bellow (Fig.2).

Fig.1 CLSM observation of emulsions stabilized with different ratios of Gli/

CAS NPs (Xu, et al, Food Structure, 33, 100285)

Fig.2 CLSM images of Pickering emulsion gel with different concentrations of kC

L210. What do authors associate with a dark shadow, it is unclear which part of Fig. 3 they are referring to. 

Reply: Thank you for your kind advice. The dark shadow is the residual oil still exists in the gel network due to the inability to remove tea oil during lyophilization, which is shown in the black shadow. We have stated in the paper.

L212. What is understood as the “change law. Please use a clearer term.  

Reply: The sentence has been revised according to your kind advice.

L213. What do the authors mean as more obvious network structure, another sentence would be preferred in this case. 

Reply: The sentence has been revised.

L220-L222. Please provide references to these observations. 

Reply: Thank you for your kind advice. We have added some more references in the discussion part.

L225. Using arrows to point at the tea oil droplets help to better understand the information conveyed. 

Reply: Thank you for your kind advice. We have added the arrows in the picutures.

L234. Please reference the previous result. 

Reply: Thank you for your kind advice. We have revised the sentence.

L240. Please provide a reference to the affirmation of the adsorption towards the interface. It is hard to identify the different structures in the image, could you please indicate which are the XG/Ly NPs in Fig. 5A-B. 

Reply: Thank you for your kind advice. We have tried to point out XG/Ly NPs in the interface. But it is very hard to explicitly indicate the particles because XG/Ly NPs is very small with dozens of nanometers. In our previous studied, we also found that Cryo-SEM images could locate it but hard to clearly emerge in the picture [1-3].

References

[1] Zhifan Li, Shuqing Zheng, Cong Zhao, Mengru Liu, Zirui Zhang, Wei Xu, Denglin Luo, Bakht Ramin Shah. Stability, microstructural and rheological properties of Pickering emulsion stabilized by xanthan gum/Lysozyme nanoparticles coupled with xanthan gum, International Journal of Biological Macromolecules, 2020, 165, 2387-2394.

[2]Wei Xu, Zhifan Li, Haomin Sun, Shuqing Zheng, He Li, Denglin Luo, Yingying Li, Mengyuan Wang, Yuntao Wang,High Internal-phase Pickering Emulsions Stabilized by Xanthan Gum/Lysozyme Nanoparticles: Rheological and Microstructural Perspective, Frontiers in Nutrition,2022, 8, 744234.

[3]Wei Xu, Zhifan Li, He, Li, Haomin Sun, Shuqing Zheng, Denglin Luo, Yingying Li, Yuntao Wang, Bakht Ramin Shah, Stabilization and microstructural network of Pickering emulsion using different xanthan gum/lysozyme nanoparticle concentrations, LWT, 2022, 160, 113298.

L230-242. The discussion can be more profound, please include a more in-depth information about the phenomena observed and the reasons that could lead to these structures to be used in food formulations. 

Reply: Thank you for your kind advice. We have provided some application in food in the according places of the paper.

L244. Please provide visual helps to better understand the structures inside Fig. 5A.B 

Reply: Thank you for your kind advice. Actually, it could not obviously observe the change between A and B. The different between the two pictures is the external network structure.

L274. It appears as if G’ and G’’ of the sample with 2 % XG/Ly and 2% kC will cross after 95 °C. Rather than being irreversible, the gel does not melt in the temperature range studied. Otherwise, if there is an irreversibility; please explain the process in more detail and provide the corresponding references. 

Reply: Thank you for your kind suggestion. As we known that the kC gel is irreversibility gel. We also provide the information and the corresponding references in the paper.

L275. In which sense the result is coherent with the reference provided. Please indicate, briefly, how the results reported in that reference can help to explain or compare to the results reported in this work. 

Reply: According to your kind suggestion, we have enriched the information of the reference and briefly introduced the research contents.

L277-279. Please elaborate a little bit on the phenomenon present in this thermal process, provide references. 

Reply: Thank you for your kind suggestion. We have revised and added some explanation in the place. Some discussion is listed below: all samples show thermal hysteresis which becomes more obvious with the increase of kC concentration. The phenomenon maybe resulted from the multiphase Pickering emulsion system and different heat-mass transfer effect. Meanwhile, kC enhanced the thermal stability of the Pickering emulsion gel and the capacity depend on XG/Ly NPs concentration.

L282. It is odd that G’ decreases at high frequencies, please verify that the system is effectively in the Lineal viscoelastic range, as this zone decreases at higher frequencies. Also tan (delta) should not be below 1, apparently the data has inertia correction problems, as G’ sometimes adopt negative values perhaps a larger geometry would avoid these issues.  

Reply: Thank you for your kind suggestion. We have also noted the phenomenon. The the Lineal viscoelastic range is confirmed in the method. The phenomenon may induced from the structure of Pickering emulsion gel was sectional destroyed.

L299-302. Please provide a reference, it is also not clear why an increase in the content of XG/Ly NPs reduces G’ of the 1 % kC samples at high frequency. 

Reply: We have revised the information of the part of paper according to your kind suggestion.

L302-320. While this comparison is interesting, please verify the limits of the Power Law equation, as it is used in the Lineal Viscoelastic Regime and this zone is not present for all the samples. Please revise and correct the analysis appropriately. It is uncommon to have negative values for tan(delta), please verify. 

Reply: Thank you for your kind suggestion. We have carried the test again. And the data were revise and correct the analysis appropriately.

L323-351. Results indicate that even at the highest level of kC, the sample can melt at longer times after 40 °C, however, this behavior is clearly affected by the presence of XG/Ly. However, it is unclear the underlying phenomena that is occurring in this case, please include more references, compare to previous works, and explain what is happening in a clearer way. 

 Reply: Thank you for your kind suggestion. While it rarely found Pickering emulsion stabilized by XG/Ly NPs. We have tried to explain in the paper. Based on our previous study, there are a certain amount of free XG and Ly in the aqueous phase. They have an interaction with kC at high 40 °C, especially for Ly. The physical interaction may make the melt at longer times.

L350-351. Please redact the line in a more technical writing style. 

 Reply: The physical interaction may make the melt at longer times. The corresponding sections have been modified.

L357. Please indicate why hardness is relevant to this work. 

 Reply: Hardness is an important parameter of Pickering emulsion gel which paly an role in the application in food.

L371-373. The formation of a 3D network of kC, which is mentioned in previous paragraphs shall not be neglected to explain hardness. Also please indicate if a finer oil droplet would affect this property. Provide references. 

Reply: Thank you for your kind suggestion. We has added the related discussion in the paper.

L387. Please supply a clearer explanation, although according to the standard deviations there is no statistically significant difference between 1 and 2% kC samples. 

Reply: It actually showed that WHC of the emulsion with a 2% kC concentration (about 99% on average) is higher than that of the Pickering emulsion with a 1% kC concentration (about 98% on average), indicating that a tighter network structure was formed inside the Pickering emulsion gel with the increase of kC concentration. But there is basically no difference in WHC of the Pickering emulsion gels with 1 and 2% kC because of their high WHC capacity.

L397. Please supply a reference. 

Reply: We have supply a reference according to your kind suggestion.

L399. Please supply a reference. 

Reply: We have supply a reference according to your kind suggestion.

L401. It would be good to do a very brief summary of the importance of these results as a way to close the results and previous to conclusions. 

Reply: Thank you for your kind suggestion. We have added a brief summary of the importance of these results.

Conclusions 

This section is brief and indicates the most important results of the work, however there is a disconnect between this part and the introduction, as emulsion stability is important in the latter, while it is clearly absent in the rest of the work. At least the introduction shall be reformulated to address this problem. 

Reply: Thank you for your kind suggestion. We have written the results of the work again which emphasize the important results briefly.

L417. The WHC value of the gels do not differ too much, but it would be good to see the actual values and their standard deviations to be sure. 

Reply: Thank you for your kind suggestion. We have revised the sentence.

Round 2

Reviewer 1 Report

Although the manuscript has been improved, especially the introduction, there are still numerous English mistakes throughout the entire manuscript which makes it hard to understand what authors mean. I suggest, as was already suggested in the first revision, that authors check the manuscript for English. As it seems no English check was carried out, as the verb tenses used in each section remains inappropriate.

Figure 1, 2, 5: Author should indicate what sample is each of the images, just saying “different concentrations of kC and XG/Ly NPs” is not acceptable. Also, why the black background? In general, all figures need to be revised.

Table 1: No standard deviation? These exp were not repeated?

Some other comments:

Line 41: “Since this” then “what”? Sentence is incomplete…

Line 45: English, sentence I not making sense

Line 48: Compared

Line 54: English, sentence I not making sense

Line 55: And what method was that?

Line 59: Gelata?

Line 83: were

Line 86, 93: English check

Lines 102-104, 124: Re-write

Line 131: Remove “typical”

Line 157: In parallel?

Line 159: It contained? Wasn’t the emulsion put in there? How many grams?

Line 184: Varied

Line 189: What is GDL?

Line 233: “I“`?. You noted that based on what? Were experiments carried out with this material?

Line 248: heading

Author Response

Dear,

Thank you for your useful comments and suggestions on our manuscript. We have modified the manuscript accordingly and the detailed corrections are listed below point by point.

Although the manuscript has been improved, especially the introduction, there are still numerous English mistakes throughout the entire manuscript which makes it hard to understand what authors mean. I suggest, as was already suggested in the first revision, that authors check the manuscript for English. As it seems no English check was carried out, as the verb tenses used in each section remains inappropriate.

Reply: Thank you for your valuable advice. We have revised the paper and very carefully checked the language. Every author have checked the language of the paper. You can checked the article again. Several times revision have been taken by authors. All the new modification has been marked in red. We hope the revision could reach your standard.

Figure 1, 2, 5: Author should indicate what sample is each of the images, just saying “different concentrations of kC and XG/Ly NPs” is not acceptable. Also, why the black background? In general, all figures need to be revised.

Reply: Thank you for your kind suggestion. We have revised the title of the Figures. We think the 2d coordinates could displayed the Pickering emulsion gel clearly. In our several published paper, we widely used the method which could clear instruct the two-factored samples [1-3]. Actually, we do not displayed the results of the black background (with our kC). There are two reasons for the deficiency. One main reason is that in our previous published papers, we have detail studied the stability, rheological properties and microstructure of Pickering emulsions stabilized with different XG/Ly NPs concentrations [1-2]. We have comparative study in the paper. The other reason is that we main researched kC-based Pickering emulsion gels in the paper. While, Pickering emulsions could not formed gel without kC.

[1] Wei Xu, Zhifan Li, He, Li, Haomin Sun, Shuqing Zheng, Denglin Luo, Yingying Li, Yuntao Wang, Bakht Ramin Shah, Stabilization and microstructural network of Pickering emulsion using different xanthan gum/lysozyme nanoparticle concentrations, LWT, 2022, 160, 113298.

[2]Wei Xu, Zhifan Li, Haomin Sun, Shuqing Zheng, He Li, Denglin Luo, Yingying Li, Mengyuan Wang, Yuntao Wang,High Internal-phase Pickering Emulsions Stabilized by Xanthan Gum/Lysozyme Nanoparticles: Rheological and Microstructural Perspective, Frontiers in Nutrition,2022, 8, 744234.

[3] Wei Xu, Shuqing Zheng, Wenjie Xi, Haomin Sun, Yuli Ning, Yin Jia,  Denglin Luo, Yingying Li, Bakht Ramin Shah, Stability, rheological properties and microstructure of Pickering emulsions stabilized by different concentration of glidian/sodium caseinate nanoparticles using konjac glucomannan as structural regulator, Food Structure, 2022, 33, 100285.

Table 1: No standard deviation? These exp were not repeated?

Reply: Thank you for your valuable advice. Each sample we tested more than three time. There is no standard deviation in the table because the data is fitted and calculated from the origin rheological data.

Some other comments:

Line 41: “Since this” then “what”? Sentence is incomplete…

Reply: According to your kind suggestion, we have revised the sentence bellow ‘Pickering emulsions stabilized by XG/Ly NPs were stable in a period of time for droplet coalescence and Ostwald ripening. Therefore, different approaches have been proposed to improve the current situation, such as polysaccharide intervention and increasing particle stabilizer concentration’.

Line 45: English, sentence I not making sense

Reply: The sentence has been revised according to your kind suggestion.

Line 48: Compared

Reply: The mistake has been revised.

Line 54: English, sentence I not making sense

Reply: The sentence has been revised as ‘The simple method of gel formation is generally used easy-gelatinized protein and polysaccharide’.

Line 55: And what method was that?

Reply: The detail information has provided in the paper. ‘For example, Li provided a simple method to construct Pickering emulsion gels based on chitosan hydrochloride-carboxymethyl starch nanogels stabilized Pickering emulsions coupled curdlan gelation’.

Line 59: Gelata?

Reply: The word has been revised.

Line 83: were

Reply: The word has been revised according to your kind suggestion.

Line 86, 93: English check

Reply: The sentences have been revised according to your kind suggestion.

Lines 102-104, 124: Re-write

Reply: The sentences have been rewritten according to your kind suggestion.

Line 131: Remove “typical”

Reply: The word has been removed.

Line 157: In parallel?

Reply: The sentence has been revised.

Line 159: It contained? Wasn’t the emulsion put in there? How many grams?

Reply: The Pickering emulsion about 25ml was put in the centrifuge tubes and then measured the WHC.

Line 184: Varied

Reply: The word has been revised.

Line 189: What is GDL?

Reply: We have change GDL with its full name (glucono-δ-lactone (GDL)) in the paper.

Line 233: “I“`?. You noted that based on what? Were experiments carried out with this material?

Reply: The mistake has been revised. It is not ‘I’, but is ‘it’ here.

Line 248: heading

Reply: The title has been revised according to your kind suggestion.

Reviewer 2 Report

The authors made substantial changes to the paper, improving its quality.

Nevertheless, there are some major issues, basically in the introduction and results and discussion sections. 

THe introduction has been improved, but its redaction shall be improved, as it is difficult to understand and tends to lose its focus; resulting in a deviation from the original message intended in the introduction.

The results have improved as well, but there still requires some comparison with previous work and references to explain better the very interesting phenomena that is being presented in this work.

The language requires important revision, and there are minor revisions that shall be addressed before it is publishable.

Introduction

The text must be reviewed, the introduction is not well written and the message is lost in the mistakes of the redaction of different sentences.

L38. Please check the language, as there is awkeard phrasing. Also please include an estimate in time of the stability, the emulsion lasted for some minutes or for some hours?

L42. I think the word “intervention” is not accurate in this context, perhaps “addition” would be better

L45. I fail to understand this sentence” Pickering emulsion gel is one phase dispersed in another enable the design of…” I think it misses a connector to make it clearer.

L48. I think the word “compared to” would suit better in this sentence

L50. Please provide a reference

L45-L61. The term Pickering emulsion is used almost 11 times, please simplify the text to make it easier to read.

L54-55. I do not understand what is meant by “The simple method is used easy-gelatinized protein and polysaccharide”

L55. Please emphasize the results obtained by the reference and focus on the type of property that the authors wish to highlight and would be relevant for the work.

L59. What is understood by gelata?

L63. Please provide a reference

L66. What is meant by “helps cooperate with colloids”? Please be more precise in the way the information is presented

L71. Perhaps the authors mean that is proposed, as the use of was implies that it was proposed in a previous work, if so, please provide the reference.

L74. Texture analysis

L77. The results will broaden or have the potential to broaden the…

Materials and methods

L79. Please revise the phrasing

L86. Please revise, it appears to be a typo in “was purchased from was obtained”

L93. The phrase “XG and Ly mixture solutions with concentration 1.0 mg/mL respective” is not easy to understand, what is meant by respective? It appears to imply as if there are more than one concentration (one for XG and one for Ly)

L99-100. The phrase “The final concentration of XG/Ly NPs and kC in the complex was 0.5%, 1%, and 2%, respectively” is no clear enough, maybe it would be better to indicate that the XG/Ly will be evaluated at concentrations of 0,5; 1 and 2 %; while the XG will be independently varied using the same concentration values.

L111-116. Is there a reason that the strain changed between the temperature (1%), time (0.5 %) and frequency sweep (0.1 %)? If so, please explain in the manuscript as it is required that for any of the three measurements the sample must be in the lineal viscoelastic region.

L137. Please include the units of 488 (nm)

Results and discussion

L182. It is not clear for me, how the polysaccharide promotes the adsorption of the particles. It is not required to be explained here but it should be explained in the manuscript.

 L191. I think the authors mean the word “equivalent”

L194. A black frame is observed behind the image, please review it

L197. A black frame is observed behind the image, please review it.

L216. Please provide a reference of the increase in viscosity, as flow measurements were not performed.

L202-203. Please correct the redaction of these lines “There is residual oil still exists…”

L213-214. Please keep in mind that the rheological measurements have not been showed at this point, so before these lines there is no experimental evidence to back up these claims. Please provide a reference or change their redaction.

L233-236. Please verify this line. I do not understand the meaning of the line “I noted that Pickering emulsion gel stabilized by octenylsuccinate quinoa starch granule with different gel networks by modulating the oil volume fraction could develop as a carrier of lutein” I think there is something missing in the sentences, that helps to improve its clarity. It would be interesting to indicate the formulation that allowed the shelf-life increment from 12 to 41 days.

L245. The black background behind the image does not allow the observation of the axes in the Figure.

L271. Please elaborate on the Results obtained by Li & Gong, in which sense they were corroborated by your results?

L273. Please revise this line, there is an awkward phrasing of some of the information.

L274-275. Please provide a reference associating this behavior to the transport phenomena present.

L276-277. Please review the redaction of these lines, its clarity can be improved.

L291. Please write Figure with a capital F. I think the authors are referring to the photographs of Figure 8.

L294. Please provide a reference to “determined to be emulsion gel from the point of view of rheology.”

L297. Please provide a reference to “That an emulsion gel system with a strong gel structure was formed”

L299. It is unclear which is the frequency where the structure collapses is it 116 Hz or 291 Hz, please reorganize the line to convey a clearer message and revise the units, as Fig. 6 depicts Pa vs rad/s

L300. Please provide a reference.

L310. Please write Figure with a capital F.

L321-338. The information is hard to understand, if it is possible to simplify this paragraph it would greatly help to improve the paper. Otherwise it can be maintained in its current form.

L351. Please indicate what is meant by high 40 °C. Also, please provide the reference about the interaction kC and the free XG or Ly.

L353. Please review the image, there is a black background that compromises its intelligibility.

L362. The sentence “…emulsions did not form emulsion gels when the concentration of XG/Ly NPs is low” is akward, please consider its redaction

L379. Please review the image, it has a black background that makes it difficult to read.

L382. The image has an abnormal shadow.

L399-400. Please check the redaction of this section.

L400. Please provide a reference.

L403. Please elaborate about this improvement of the tea oil.

L384-403. Please compare these results of water retention with previous literature, it would be interesting to see how the results present in this work relate to other Pickering emulsion gels or emulsion gels in general. 

Conclusions.

L405. Please revise the sentence “from sol to gel state based self-gelation of kC from microstructure”, as it could be improved in its clarity.

Author Response

The authors made substantial changes to the paper, improving its quality.

Nevertheless, there are some major issues, basically in the introduction and results and discussion sections. The introduction has been improved, but its redaction shall be improved, as it is difficult to understand and tends to lose its focus; resulting in a deviation from the original message intended in the introduction.

Reply: Thank you for your valuable advice. We have revised the paper and very carefully and revised the other part of the paper. We have revised the introduction part according to your suggestion. While we introduce the stability of the Pickering emulsion is just to introduce the job further studied based our previous research work. It is a common sense that Pickering emulsion gel has excellent stability compared with Pickering emulsion. As the title displayed that the paper is just discuss carrageenan-based Pickering emulsion gels from microstructure, rheological, and texture perspective. According to your kind suggestion, we have also revised the results and discussion sections.

The results have improved as well, but there still requires some comparison with previous work and references to explain better the very interesting phenomena that is being presented in this work.

Reply: Thank you for your valuable advice. We have compared with previous work and references to explain the results.

The language requires important revision, and there are minor revisions that shall be addressed before it is publishable.

Reply: Thank you for your valuable advice. We have revised the paper and very carefully checked the language. Every author have checked the language of the paper. You can checked the article again. Several times revision have been taken by authors. All the new modification has been marked in red. We hope the revision could reach your standard.

Introduction

The text must be reviewed, the introduction is not well written and the message is lost in the mistakes of the redaction of different sentences.

Reply: According to your suggestion, we try to revise the paper as we can in the 5 days for modification. We hope the revision could reach your standard.

L38. Please check the language, as there is awkeard phrasing. Also please include an estimate in time of the stability, the emulsion lasted for some minutes or for some hours?

Reply: We checked the expression carefully and revised the sentence.

L42. I think the word “intervention” is not accurate in this context, perhaps “addition” would be better

Reply: We have revised the sentence according to your advice.

L45. I fail to understand this sentence” Pickering emulsion gel is one phase dispersed in another enable the design of…” I think it misses a connector to make it clearer.

Reply: Thanks for your advice. We have revised the sentence as ‘Pickering emulsion gel is two-phase systems of which one phase dispersed in another, where the microstructures and viscoelastic properties can be tailored’.

L48. I think the word “compared to” would suit better in this sentence

Reply: The word has changed.

L50. Please provide a reference

Reply: We have added a reference in the appropriate place.

L45-L61. The term Pickering emulsion is used almost 11 times, please simplify the text to make it easier to read.

Reply: We have simplify the text simplify the text.

L54-55. I do not understand what is meant by “The simple method is used easy-gelatinized protein and polysaccharide”

Reply: The sentence has been revised according to your advice.

L55. Please emphasize the results obtained by the reference and focus on the type of property that the authors wish to highlight and would be relevant for the work.

Reply: We used the reference is to emphasize gel formation could use polysaccharide gelatinization. The reference is relevant for the work and the content.

L59. What is understood by gelata?

Reply: The mistake has been revised.

L63. Please provide a reference

Reply: We have added a reference in the paper.

L66. What is meant by “helps cooperate with colloids”? Please be more precise in the way the information is presented

Reply: We have revised the sentence as according to your advice.

L71. Perhaps the authors mean that is proposed, as the use of was implies that it was proposed in a previous work, if so, please provide the reference.

Reply: The sentence has been revised and the the reference was provided.

L74. Texture analysis

Reply: The sentence has been revised.

L77. The results will broaden or have the potential to broaden the…

Reply: The sentence has been revised according to your kind suggestion.

Materials and methods

L79. Please revise the phrasing

Reply: We have checked the phrasing.

L86. Please revise, it appears to be a typo in “was purchased from was obtained”

Reply: We have checked the sentence and revised the expression.

L93. The phrase “XG and Ly mixture solutions with concentration 1.0 mg/mL respective” is not easy to understand, what is meant by respective? It appears to imply as if there are more than one concentration (one for XG and one for Ly)

Reply: According to your kind suggestion, we have checked the sentence and revised the expression. The detail preparation method of XG/Ly NPs has been stated in the 2.2.

L99-100. The phrase “The final concentration of XG/Ly NPs and kC in the complex was 0.5%, 1%, and 2%, respectively” is no clear enough, maybe it would be better to indicate that the XG/Ly will be evaluated at concentrations of 0,5; 1 and 2 %; while the XG will be independently varied using the same concentration values.

Reply: According to your kind suggestion, we have revise the appropriate place in the paper.

L111-116. Is there a reason that the strain changed between the temperature (1%), time (0.5 %) and frequency sweep (0.1 %)? If so, please explain in the manuscript as it is required that for any of the three measurements the sample must be in the lineal viscoelastic region.

Reply: We have checked the information again. We used in 1% and 0.5% in the paper which both is within the lineal viscoelastic region

L137. Please include the units of 488 (nm)

Reply: The mistake has been revised.

Results and discussion

L182. It is not clear for me, how the polysaccharide promotes the adsorption of the particles. It is not required to be explained here but it should be explained in the manuscript.

Reply: Thank for your kind suggestion. We have removed the statement according to your suggestion.

 L191. I think the authors mean the word “equivalent”

Reply: The word has been changed.

L194. A black frame is observed behind the image, please review it

Reply: We checked it again.

L197. A black frame is observed behind the image, please review it.

Reply: We checked it again.

L216. Please provide a reference of the increase in viscosity, as flow measurements were not performed.

Reply: We provide a reference here according to your kind suggestion.

L202-203. Please correct the redaction of these lines “There is residual oil still exists…”

Reply: We have revised the sentence according to your kind suggestion.

L213-214. Please keep in mind that the rheological measurements have not been showed at this point, so before these lines there is no experimental evidence to back up these claims. Please provide a reference or change their redaction.

Reply: According to your kind suggestion, we have revised content. We change the statement as ‘Addition of polysaccharides may promote’.

L233-236. Please verify this line. I do not understand the meaning of the line “I noted that Pickering emulsion gel stabilized by octenylsuccinate quinoa starch granule with different gel networks by modulating the oil volume fraction could develop as a carrier of lutein” I think there is something missing in the sentences, that helps to improve its clarity. It would be interesting to indicate the formulation that allowed the shelf-life increment from 12 to 41 days.

Reply: According to your kind suggestion, the sentence has been revised.

L245. The black background behind the image does not allow the observation of the axes in the Figure.

Reply: We checked the Figure again.

L271. Please elaborate on the Results obtained by Li & Gong, in which sense they were corroborated by your results?

Reply: We have removed the sentence here because the reference has been discussed previously in the paper.

L273. Please revise this line, there is an awkward phrasing of some of the information.

Reply: We have revised the sentence.

L274-275. Please provide a reference associating this behavior to the transport phenomena present.

Reply: Thanks for your kind suggestion. We have checked the related content in the paper. Here we used maybe to explain the phenomena.

L276-277. Please review the redaction of these lines, its clarity can be improved.

Reply: We have checked the content. Thanks for your kind suggestion.

L291. Please write Figure with a capital F. I think the authors are referring to the photographs of Figure 8.

Reply: We have revised the content.

L294. Please provide a reference to “determined to be emulsion gel from the point of view of rheology.”

Reply: Thanks for your kind suggestion. We have removed the related content in the paper.

L297. Please provide a reference to “That an emulsion gel system with a strong gel structure was formed”

Reply: We have provided a reference here to support the viewpoint.

L299. It is unclear which is the frequency where the structure collapses is it 116 Hz or 291 Hz, please reorganize the line to convey a clearer message and revise the units, as Fig. 6 depicts Pa vs rad/s

Reply: Thanks for your kind suggestion. The related information has been revised in the paper.

L300. Please provide a reference.

Reply: Thanks for your kind suggestion. Here we use ‘may’ which we try to explain the phenomenon.

L310. Please write Figure with a capital F.

Reply: We have checked all the paper and revised as Fig.

L321-338. The information is hard to understand, if it is possible to simplify this paragraph it would greatly help to improve the paper. Otherwise it can be maintained in its current form.

Reply: Thanks for your kind suggestion.

L351. Please indicate what is meant by high 40 °C. Also, please provide the reference about the interaction kC and the free XG or Ly.

Reply: We have revised the paper for easy readability. Thanks for your kind suggestion.

L353. Please review the image, there is a black background that compromises its intelligibility.

Reply: We have checked the image and we were puzzled with the background.

L362. The sentence “…emulsions did not form emulsion gels when the concentration of XG/Ly NPs is low” is akward, please consider its redaction

Reply: Thanks for your kind suggestion. We have revised the sentence.

L379. Please review the image, it has a black background that makes it difficult to read.

Reply: We have checked the image.

L382. The image has an abnormal shadow.

Reply: We have checked the image and we were puzzled with the background.

L399-400. Please check the redaction of this section.

Reply: Thanks for your kind suggestion. We have revised the sentence.

L400. Please provide a reference.

Reply: We have revised the statement.

L403. Please elaborate about this improvement of the tea oil.

Reply: We have revised the sentence.

L384-403. Please compare these results of water retention with previous literature, it would be interesting to see how the results present in this work relate to other Pickering emulsion gels or emulsion gels in general.

Reply: Thanks for your kind suggestion. We have tried to compare these results of water retention with previous literature.

 Conclusions.

L405. Please revise the sentence “from sol to gel state based self-gelation of kC from microstructure”, as it could be improved in its clarity.

Reply: Thanks for your kind suggestion. We have rewritten the sentence.
